



**A DOM continuum from the roof of the world – Tibetan molecular dissolved organic**
**matter characteristics track sources, land use effects, and processing along the**
**fluvial-limnic pathway**
Philipp **Maurischat** [1,2], Michael **Seidel** [3], Thorsten **Dittmar** [3,4], and Georg **Guggenberger** [1]
[1] Leibniz University Hannover, Institute of Soil Science, 30419 Hannover, Germany
[2] Institute of Biology and Environmental Sciences (IBU), Carl von Ossietzky University of Oldenburg,
26129 Oldenburg, Germany
[3] Institute for Chemistry and Biology of the Marine Environment (ICBM), Carl von Ossietzky University
of Oldenburg, 26129 Oldenburg, Germany
[4] Helmholtz Institute for Functional Marine Biodiversity at the University of Oldenburg (HIFMB),
26129 Oldenburg, Germany
**Correspondence to: Philipp Maurischat (maurischat@ifbk.uni-hannover.de)**
**Keywords:** Non-metric multidimensional scaling (NMDS), Alpine pastures, Natural organic matter
(NOM), Land use controls, Molecular NOM composition, Fourier-transform ion cyclotron resonance
mass spectrometry (FT-ICR-MS)
**Funding:** This research is a contribution to the International Research Training Group "Geo-
ecosystems in transition on the Tibetan Plateau (TransTiP)", funded by Deutsche
Forschungsgemeinschaft (DFG grant 317513741 / GRK 2309). Funding for MS by Cluster of Excellence
EXC 2077 "The Ocean Floor – Earth's Uncharted Interface" (Project number 390741603).
**Conflict of interest:** No conflict of interest.



## Abstract

The Tibetan Plateau (TP) is the world largest and highest plateau, also comprising the biggest
connected alpine pasture system of the world. Like other alpine systems, it is sensitive to impacts by
climate change and increasing anthropogenic pressure. Carbon cycling at the TP is complex, including
sources such as primary production in lakes, glaciers, and terrestrial plants, agricultural land use but
also organic matter (OM) from aeolian deposition. Dissolved organic matter (DOM) connects these
carbon reservoirs in the network, following the hydrological cycle from precipitation, glaciers, and
headwaters to lakes. DOM is highly complex, its molecular composition holds information from its
diverse sources and transformations during transport. However, due to its complexity, DOM cycling
along the headwater-fluvial-limnic pathway and how terrestrial change can impact carbon cycling in
the diverse water bodies is still not well understood. Here, we study DOM molecular transformations
using ultrahigh-resolution mass spectrometry (FT-ICR-MS) along the TP alpine continuum from
glacial, groundwater springs, and wetland biomes including pastures and alpine steppe, to the large
saline endorheic Lake Nam Co. DOM molecular composition differed with respect to allochthonous
sources between endmembers, as well as between stream samples, the brackish mixing zone, and
the lake. Glacial meltwater DOM contained autochthonous signatures of low-oxidised, unsaturated
molecular formulae together with terrestrial-like, dust-borne DOM sources. Glacial-fed streams were
characterised by fresh autochthonous, probably algal DOM, and aromatic compounds likely
originating from pastoral land sources. DOM from a groundwater spring had a highly degraded,
strongly oxidised signature, probably related to the shallow upper aquifer, and degraded pastoral
land sources. Wetland and stream DOM were characterised by less oxidised and less degraded inputs
from vascular plants and soils. At the brackish zone of the lake shore, DOM contained a mixture of
lake- and terrestrial DOM inherited from the streams. At Lake Nam Co, depletion of aromatic
terrestrial molecular formulae suggested photooxidation at the surface, and relative enrichment of
potentially recalcitrant DOM within the lake. Additionally, a relative enrichment of more aliphatic,
nitrogen-containing DOM suggests autochthonous algal and microbial DOM sources in the lake. Our
study revealed that DOM composition was largely influenced by local sources and transformations in
glaciers, wetlands, and groundwater springs, also incorporating molecular signatures of pasture
degradation. Streams with less glacial influence had plant- and soil borne aromatic-rich DOM sources,
while the endorheic Lake Nam Co had a recalcitrant DOM composition comparable to millennial-
scale stable marine DOM. This suggests that there is no typical high-alpine DOM signature, but that
complex processes form DOM characteristics in the fluvial-limnic continuum. Small-scale catchment
properties, land degradation and aquatic domains shape the differences. Alpine DOM compositions
hence appear to be closely linked to landscape properties suggesting their susceptibility to changes in
water quality and OM cycling in sensitive High Asian ecosystems.



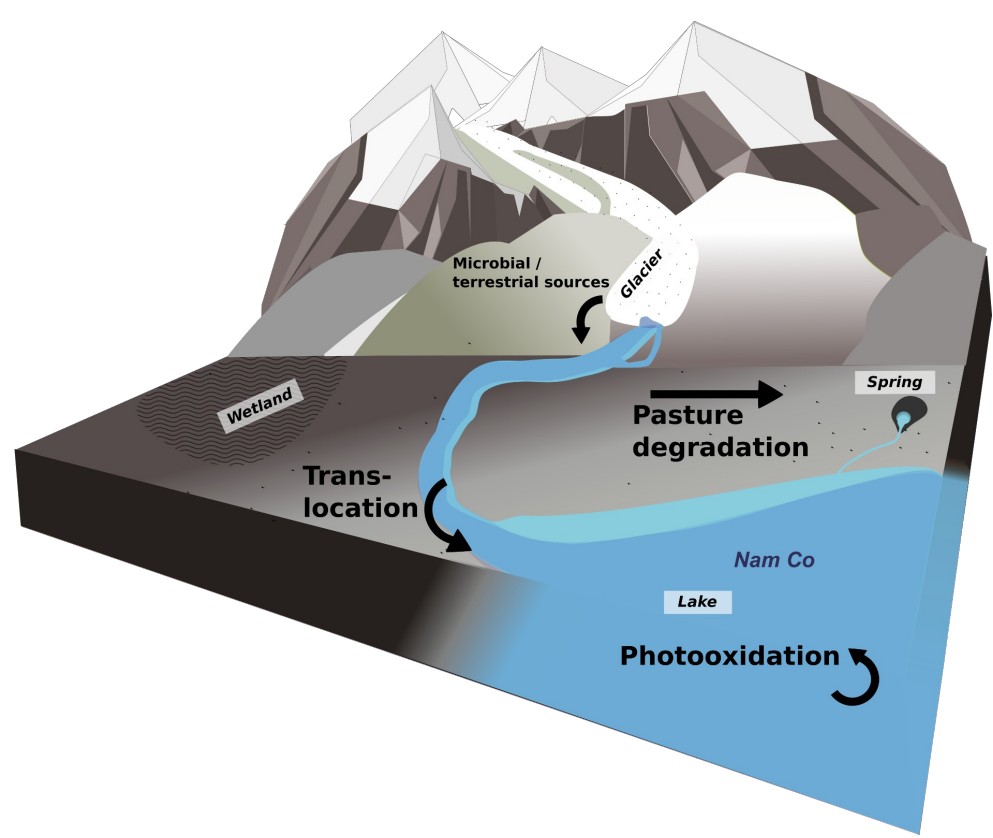

**Graphical Abstract: Main processes shaping DOM molecular characteristics and transformation in**
**the high-alpine Nam Co watershed.**
**1. Introduction**
The Tibetan Plateau (TP) comprises the largest alpine pasture system of the world (Miehe et al.,
2019). It is also known as Asia's water tower (Bandyopadhyay, 2013), forming the source of many
large river systems, providing freshwater resources to billions of people in East and South-East Asia.
Large quantities of the water are stored in the ice masses of the TP, forming the largest frozen
freshwater reservoir outside the Polar Regions. This third pole environment is well investigated (Qiu,
2008; Yao et al., 2012), revealing that High Asia's ecosystems are threatened (Hopping et al., 2018)
by climate change (Yao et al., 2019) and by intensification and other changes of land use (Harris,
2010). Emerging freshwater quality issues (Qu et al., 2019) and the unknown consequences of peak
water (McDowell et al., 2022) highlight the utmost importance of the TP for up- and downstream
societies. Nutrient and energy cycles of ecosystems are connected by dissolved organic matter
(DOM) fluxes (Spencer et al., 2014). Biogeochemical processes in the source area shape the
molecular composition of DOM (Liu et al., 2020; Roebuck et al., 2020; Seifert et al., 2016; Wilson and
Xenopoulos, 2009). We hypothesize that DOM molecular characteristics entail markers for climatic



and land use changes on the TP and that these markers can be used to differentiate between
landscape units and endmembers. Understanding the marker function of DOM will allow to better
foresee carbon processing at present and under changing environmental conditions. Lastly, this will
be key to assess the potential vulnerability and responses of high alpine ecosystems to the challenges
of the Anthropocene.
DOM connects fluvial ecosystems over several hundreds of kilometres (Seidel et al., 2015), and links
terrestrial and aquatic biomes (Goodman et al., 2011). Changes in catchment properties might
therefore trigger effects in distanced, fragile ecosystems (Wilson and Xenopoulos, 2009), showing
that action and response of land use or climatic changes can be temporally and spatially detached
(Goes et al., 2020; Roulet and Moore, 2006). The molecular diversity and complexity of DOM calls for
the employment of advanced techniques (Fellman et al., 2010; Nebbioso and Piccolo, 2013). This led
to analytical advancements such as electrospray ionization Fourier-transform ion cyclotron
resonance mass spectrometry (FT-ICR-MS), an ultrahigh-resolution mass spectrometry method
allowing to identify thousands of molecular formulae in samples of natural organic matter (NOM),
thus offering a unique opportunity to understand molecular DOM characteristics, sources and
transformations (Hawkes et al., 2020).
Here, we used FT-ICR-MS to gain insight about molecular information and elemental composition of
DOM, to understand characteristics and processing of stream DOM of different catchments, glacier,
groundwater spring and alpine wetlands endmembers and DOM of an endorheic lake. The Nam Co
watershed, located in the southern/central part of the TP lies in the transition zone of the *Kobresia*
*pygmaea (K. pygmaea)* dominated alpine pasture biome (Miehe et al., 2008) and the alpine steppe.
The area's unique positioning in this transition zone is expressed between the south, with a more
humid, glacial-influenced high-mountain ecosystem contrasting the hilly northern margin of the
catchment, with more arid climate and alpine steppe dominated vegetation, where clear signs of
pasture degradation are visible (Maurischat et al., 2022). DOM characteristics and transformation in
this complex natural interplay were only investigated to limited extent. Spencer et al. (2014) found
complex OM sources in glaciers, streams and Lake Nam Co, while a recent study highlighted
transformation of DOM in pro-glacial streams (Li et al., 2021). This leaves important questions of 1)
how DOM signatures are influenced by the diverse biotic and abiotic inventories of the watershed
with differing degree of glaciation, alpine wetlands, and groundwater sources as well as land
degradation, and the alpine pasture alpine steppe ecotone, and 2) how DOM is processed during the
fluvial pathway and in the lake. The Nam Co watershed is a sentinel for changes and a natural
laboratory (Anslan et al., 2020), suited to test whether DOM characteristics can be employed as a
precursor for changes in this sensitive alpine environment, representative for larger parts of the TP.



In this context we hypothesized: I) Catchments of the Nam Co watershed, having different DOM
sources, differ in their molecular composition of DOM, where the effect of intensified land use and
pasture degradation drives a degraded, terrestrial signature rich in recalcitrant molecular formulae.
II) The endmembers (glacial effluents, groundwater springs and alpine wetlands) along the glacier-to-
lake continuum possess unique DOM signatures compared to the integrated DOM of streams.
III) DOM transformations along the stream path are limited by cold water temperatures, large
discharge, high turbidity, and short residence times in the stream. In contrast, signatures of DOM
transformations in the clear and deep lake are expected to be dominant.
In consequence, IV) DOM signatures of lake water are chemically distinct compared to the terrestrial
sources and reveal photodegradation and biological utilisation induced processing of DOM.
## 2. Materials and methods
### 2.1 Site description and sampling
The Nam Co watershed has a total size of 10789 km$^2$. Two main landscape units can be distinguished,
the southern mountainous and the northern upland zone. The south of the watershed is
characterised by the Nyainqentanglha mountain ridge (NMR) with highest elevations of more than
7000 m. asl., which are glaciated (Bolch et al., 2010). Glacial meltwater is the dominating water
source of the streams here (Adnan et al., 2019b). Sparse vegetation dominates the glacial zone, while
*K. pygmaea* pastures are being developed at lower elevations (Miehe et al., 2019). Close to the lake
shore and on lake terraces, alpine steppe vegetation is developed (Nieberding et al., 2021). The
southern part of the watershed is characterised by higher precipitation compared to the north. Up to
530 mm y$^{-1}$ are measured in the NMR (Anslan et al., 2020). In the northern margin hilly uplands a less
steep relief dominates (Yu et al., 2021), lower annual precipitation of around 300 mm y$^{-1}$ are reported
(Anslan et al., 2020). Alpine pasture is developed on north-exposed hill-flanks and in the valley
bottoms, while alpine steppe grows on south exposed flanks, in the upland, and at the lake shoreline
(Maurischat et al., 2022). Along with the aridtiy gradient from south to north a  degradation gradient
occurs (Anslan et al., 2020). The climate of the watershed is biannual. The summer months are
dominated by the Indian Summer Monsoon from May until early September (Chen et al., 2019).
More than 80% of the annual precipitation falls during the monsoon season with mean day-time air
temperature up to 11°C (Chen et al., 2019). During winter and spring, the climate is dry and cold with
minima of around -20°C between December and February and fewer precipitation (Nieberding et al.,
2021). The endorheic Lake Nam Co with an elevation of 4726 m. asl. has a total size of 2000 km². The
lake is dimictic, oligotrophic, and lightly saline with 0.9 g l$^{-1}$ (Keil et al., 2010), and has a maximum
depth of 99 m. It is well supplied with oxygen and has a clear water column (Wang et al., 2020).



Three catchments of the Nam Co watershed were selected to represent its natural diversity (Fig. 1 a, b). The Niyaqu catchment (sample IDs 1:n in Fig. 1a) in the east has a total area of 406 km². Two streams drain the catchment, the southern stream receives meltwater from a glacial outcrop of the NMR located 700 m above lake level. This river runs through extensive alpine pastures and feeds a large alpine wetland (Fig. 1b). The northern stream drains a hilly upland area in the transition of alpine steppe and alpine pasture. Herding of yak takes place throughout the year. The Zhagu catchment (sample IDs 3:n Fig. 1a) in the arid north of the watershed is the smallest investigated catchment. It has a size of 46 km² and is mostly characterised by hilly upland relief (Keil et al., 2010). There is no glacial influence and only a small altitudinal gradient in this catchment, with the highest elevation at 5230 m. asl. Two creeks drain the catchment, both fed by groundwater springs. During investigation the catchment was arheic, clear signs of degradation of *K. pygmaea* pastures were visible (Maurischat et al., 2022), alongside alpine steppe is developed and also used for animal husbandry. In the south, the Qugaqie catchment (sample IDs 2:n Fig. 1b) represents the high-altitude zone of the NMR. The catchment has a size of 58 km² and is characterised by steep relief and a valley course in south-north direction (Keil et al., 2010). The altitudinal difference between lake and summit is 2200 m. This catchment is used as summer pasture along the stream that drains the catchment.

Water samples were taken in September 2019 following the stream paths from the source until the terminus. Potential endmembers were sampled directly, identified by following the stream discharge routes upstream.

We identified three endmember groups (glacial effluents, groundwater springs, and alpine wetlands) and three additional sampling units (stream water, brackish water, and lake water), resulting in six sample categories (Fig. 1). Glacial effluents were drawn directly at or close to the glacial terminus, while groundwater samples were drawn directly at springs. Alpine wetland samples were taken from the standing water column of areas characterised by bogs and aquatic plants. Brackish water samples were taken in the mixing zone of stream and lake water at the stream mouth at the shoreline of Lake Nam Co. Lake samples were taken offshore. Details on the characteristics of the catchments and the sampling can be obtained from Maurischat et al. (2022).



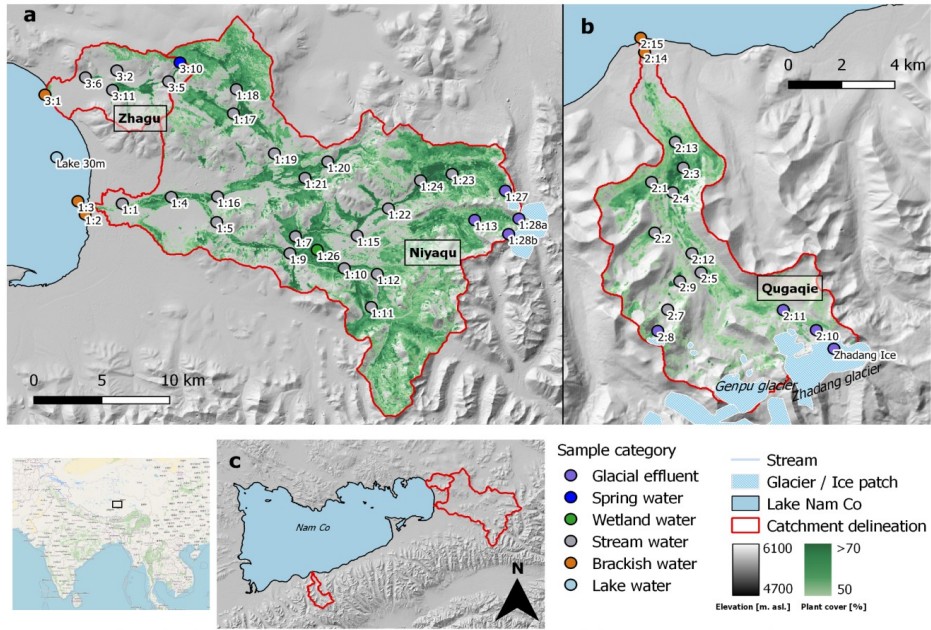

169

**Figure 1: Overview map of the investigated catchments and sampling locations with sample categories. Plant cover estimations from Maurischat et al. (2022.) (a+b) represent *K. pygmaea* pastures, the zones of most prominent yak grazing. © OpenStreetMap contributors 2022. Distributed under the Open Data Commons Open Database License (ODbL) v1.0.**

Samples were taken from the middle of the stream profile using a telescopic sampling device. Lake waters were sampled from 30 m depth with a submersible sampling device. All water samples were taken in seven subsamples with a volume of one litre each, mixed and a 500 mL aliquot of this was taken for final processing. Samples were filtered on-site using a 0.45 µm mesh size polyethersulfone membrane (Supor, Pall, Port Washington, USA), a vacuum filtration device and a portable electric pump. Samples were stored in pre-cleaned high-density polyethylene bottles (Rotilabo, Carl Roth, Karlsruhe, Germany) and kept at -21°C until analysis.

### 2.2 Solid phase extraction

Filtered DOM samples were acidified to pH 2 using 32 % HCl (Rotipuran p.a., Carl Roth, Karlsruhe, Germany). Dissolved organic carbon (DOC) concentrations were measured from 20 mL of sample by high-temperature oxidation on a total organic analyser (varioTOC Cube, Elementar, Langenselbold, Germany). DOM samples were then diluted with ultrapure water to a concentration of 1.5 mg C L$^{-1}$, and 250 mL of diluted sample was used for extraction. Cartridges with 100 mg of styrene divinyl benzene polymer (PPL) resin (Bond Elut, Agilent Technologies, Santa Clara, USA) were used for extraction. Solid phase extracted (SPE) DOM was prepared following Dittmar et al. (2008). The SPE



elute was transferred in cauterized brown glass flasks (Neochrom, Neolab Migge, Heidelberg,
Germany) sealed with polytetrafluorethylen caps (Neochrom, Neolab Migge, Heidelberg, Germany)
and stored at -18°C until further analysis. Extraction efficiency was assessed by drying 0.2 mL of SPE-
DOM under an $N_2$ atmosphere and resolving the aliquot in 20 mL of ultrapure water. The samples
were then analysed for carbon concentration by high-temperature oxidation (varioTOC Cube,
Elementar, Langenselbold, Germany) and the volumetric proportion of initial DOC to extracted SPE-
DOC was calculated. Reference material (Suwannee River / IHSS) (Green et al., 2015) was compared
to routine assays. Blank samples with ultrapure water were used to check process-induced
contamination.
2.3 Fourier-Transform Ion Cyclotron Resonance Mass Spectrometry
SPE-DOM samples were diluted in 1:1 methanol/ultrapure water to a final concentration of 5 mg C L$^{-1}$
for analysis. Samples were analysed on a SolariX XR 15 Tesla FT-ICR-MS (Bruker Daltonik, Bremen,
Germany). Duplicate sample electrospray ionization (ESI) was carried out in negative mode at a flow
rate of 122 μL h$^{-1}$. 200 broadband scans (masses of 92.14 to 2000 Da) were acquired per sample,
accumulation time was 0.2 s per scan. Internal calibration standards from the Northern Equatorial
Pacific Intermediate Water (NEqPIW) (Green et al., 2015) were used to assign mass spectra to mass
error < 0.1 ppm. Data processing followed routines of ICBM-OCEAN (Merder et al., 2020). Method
detection limits (MDL) were applied (Riedel and Dittmar, 2014) with the ICBM-OCEAN default
elemental composition and a minimum signal-to-noise ratio (S/MDL) of 2.5. Minimum signal to MDL
ratio as backbone for recalibration was 5 using mean recalibration mode. Molecular formulae were
assigned with a tolerance of 0.5 ppm as $C_{1-100}H_{1-125}O_{1-40}N_{0-4}S_{0-2}P_{0-1}$ in the mass range 95 to 1000 Da.
Molecular formulae assignments were accepted if they were present in more than 5% of the
samples. Contaminants were identified and excluded using the contaminant reference mass list.
Detection limits for peaks were normalised to sample peak intensities. Overall peak intensities were
scaled to the local sample maxima by using the sum of peaks. Isotope ($^{13}C$, $^{18}O$, $^{15}N$, $^{34}S$) mass effects
were corrected to the naturally most abundant form of each isotopologue. Molecular formulae with
molar ratios of oxygen-to-carbon (O:C) = 0, O:C ≥ 1, and hydrogen-to-carbon (H:C) > 2.5 were
removed. Duplicate samples were normalised, molecular formulae were retained only, when present
in both duplicates. NEqPIW DOM was used as an in-line calibration to measure instrument drift.
**2.4 Molecular descriptive classes, counts and indices**
Molecular formulae were assigned to molecular compound classes (Leyva et al., 2020). The original
compound classification was suggested by Šantl-Temkiv et al. (2013), compound class names were
modified according to the descriptive classes used by ICBM-OCEAN (Merder et al., 2020). The



modified aromaticity index (AI$_{mod}$) and double bond equivalents index (DBE), representing aromaticity
and unsaturation, respectively in DOM, were calculated for each formula as proposed by Koch and
Dittmar (2006; 2016). AI$_{mod}$ indices > 0.5 were assigned as aromatic, while indices ≥ 0.67 were
considered as condensed aromatic compounds. The degradation index (I$_{Deg}$) was calculated as a
measure of degradation state of OM (Flerus et al., 2012) and the terrestrial index (I$_{Terr}$) was
calculated, as a measure of terrestrial DOM sources (Medeiros et al., 2016). We further assessed the
island of stability (I$_{O}$S) (Lechtenfeld et al., 2014) to gain insights into the relative share of recalcitrant,
i.e. millennial-scale stable molecular compounds. The CHO index was calculated as a measure of
organic carbon oxidation stage (Mann et al., 2015). Low CHO values indicate highly reduced
(relatively low O content) and high CHO values indicate highly oxidized (relatively high O content)
molecular formulae. Molecular diversity was interpreted as α-diversity by depicting the intra-
community molecular diversity (Thukral, 2017), here we use the total number of molecular formulae
on catchment or endmember scale as a representative.

### 2.5 Environmental variables

Relative DOM fluorescence (FDOM) as the product of co-correlated microbial-like and terrestrial-like
FDOM, DOC, dissolved inorganic carbon concentration (DIC), δ $^{13}$C of DOC, mean plant cover at the
sampling point and the concentration of nitrate (NO$_3^-$) were taken as environmental predictor
variables from Maurischat et al. (2022) and tested for statistical correlations with the molecular DOM
data.

### 2.6 Statistics

Molecular formula intensities were rescaled between 0 and 1 and expressed in percentage. The
analysis was conducted by grouping with two independent factors. 1) The three catchments: Niyaqu,
Qugaqie, Zhagu and Lake Nam Co and 2) sample categories: endmember water (glacial effluents,
spring, and wetland) as well as site groups (stream water, brackish water, and lake water). Two
endmember groups, spring, and wetland, as well as the site group Lake Nam Co was excluded from
statistical analysis due to small sample size. Intensity weighted arithmetic means and standard
deviation were calculated for AI$_{mod}$, number of formulae with AI$_{mod}$ >0.5, number of formulae with
AI$_{mod}$ >0.67, DBE, and number of formulae containing heteroatoms nitrogen (N), phosphorous (P) and
sulphur (S), the total number of assigned formulae and for compound classes.
Data were tested for normal distribution and homoscedasticity by application of Kolmogorov-
Smirnoff test and Levene test (Brown and Forsythe, 1974). Due to violations of normal distribution in
combination with unbalanced sample sizes per group, parametrical tests were considered unreliable
(Bortz and Schuster, 2010). Multiple pairwise comparisons were conducted using Kruskal Wallis and



Mann Whitney tests (Birnbaum, 1956) for pairwise comparisons with Bonferroni post-hoc correction
for multiple testing. Significance levels ($\alpha$) of 0.05 were set as threshold (Supplementary material,
Table S1 and Table S2).
Non-metric multidimensional scaling (NMDS) was used for dimensionality reduction and ordination
(Anderson et al., 2006; Faith et al., 1987). Data for NMDS were scaled and mean-centred (Jolliffe,
2002). NMDS was conducted for both independent factors (site and sample category), while sites and
sample categories with low statistical power ($n < 3$) were not incorporated to avoid confounding.
The Bray-Curtis dissimilarity index with k = 3 was used. Loadings, scores, and $R^2$-coefficients of
determination are provided in the supplementary material (Table S3 – S7). The following compound
classes were combined because of co-correlation: aromatic O-rich and aromatic O-poor = ARO, highly
unsaturated O-rich and highly unsaturated O-poor = HUSAT, unsaturated O-rich, unsaturated O-poor
and unsaturated with N = USAT (Fig. 4). Collinear external predictor variables were removed from the
NMDS. R software (The R project for statistical computing, v3.6.3, GNU free software) was used for
statistics. The R base packages (R Core Team, 2013) and *tidyverse* (Wickham et al., 2019) were used
for data organisation and pre-processing, as well as non-parametric statistics. The packages *ggplot2*
(Wickham et al., 2019) and *vegan* (Oksanen et al., 2020) were used for graphical illustration and for
NMDS analysis, respectively.
## 3. Results
### 3.1 Sample treatment
SPE-DOM extraction efficiencies were 61.4 % ± 18.6 % (Supplementary material, Table S8). There was
no sufficient statistical explanatory power between the extraction efficiency and the retrieved
number of molecular formulae of the samples ($R^2$=.08, $F_{(1, 43)}$ =5.144, $\beta$=-0.007, p=0.02) to expect
systematic methodological failure.
### 3.2 Group counts and statistics
Samples were grouped by two independent factors, site (the three investigated catchments Niyaqu,
Qugaqie, Zhagu, or the Nam Co Lake) and sample category (the three endmembers (glacial,
groundwater spring and wetland waters) as well as stream, brackish and lake water samples). For
sites, the total number of assigned molecular formulae decreased in the order Qugaqie > Zhagu >
Niyaqu > Lake Nam Co, latter had 50 % less assigned formulae compared to the catchments (Table 1).
Water samples from wetland and brackish environments had highest numbers of assigned formulae,
followed by glacial effluents, spring and stream water, while lake water samples had the lowest
molecular diversity (Table 2).



Heteroatom-containing molecular formulae (N, S, P) had different relative abundances in sites and
sample categories. Lake Nam Co had relatively more N-heteroatoms than the three catchments
(Table 2). The relative abundance of sulphur containing formulae significantly decreased in Niyaqu
compared to Qugaqie (p=0.005), lowest mean values were found in DOM of Zhagu and Lake Nam Co.
P-containing molecular formulae were enriched in DOM of the lake water relative to DOM of the
three catchments (Table 1, Table S1).
N-containing molecular formulae were most abundant in wetland water and brackish water samples.
Spring, stream, and glacial effluents formed a group of medium N distribution and lake samples had
the lowest count of formulae associated with N. DOM and glacial DOM had the highest relative
abundances of S-containing molecular formulae, followed by wetland and stream DOM with lower
amounts, while groundwater springs and lake DOM had the lowest number of molecular formulae
associated with S. Molecular formulae containing P were least abundant in glacial and groundwater
spring DOM, followed by the order stream < lake < brackish water < wetland DOM. A high abundance
of P was visible in lake DOM at relative numbers (Table 2).
DOM of Lake Nam Co had lowest $AI_{mod}$, DBE and $I_{Terr}$ values, while DOM from Niyaqu, Qugaqie and
Zhagu exhibited no differences (Table 1). For sample categories, brackish DOM had higher $I_{Terr}$ and
DBE values but no significant differences were visible between DOM of terrestrial systems (Table 2).
No significant differences were visible for O/C ratios, but the H/C ratios were highest in Lake DOM
(Table 1). Significantly higher relative numbers of hydrogen in samples of Qugaqie were found
compared to Niyaqu (p=0.005). Additionally, the CHO index showed that Lake DOM and DOM of the
Qugaqie catchment were significantly less oxidized compared to DOM of the degraded, arheic Zhagu
catchment (p=0.042). Correspondingly, $I_{Deg}$ values, indicating relatively more degraded DOM, were
significantly higher for Zhagu compared to both the Niyaqu and Qugaqie catchment DOM (p=0.0002).
DOM of Lake Nam Co had Lowest values of $I_{Deg}$. IoS values, indicating recalcitrant DOM, showed a
significantly higher contribution in Zhagu and Lake Nam Co compared to Niyaqu and Qugaqie
(p=0.04, 0.05, respectively). In the sample categories (Table 2). H/C ratios were higher in glacial and
lake DOM and lower in wetland DOM. The CHO index showed less oxidized DOM originating from
glacial effluents and the lake compared to terrestrial sources (Fig. 5). $I_{Deg}$ values were lower in the
lake compared to glacial, stream and wetland DOM. Highest $I_{Deg}$ values were observed in DOM in the
brackish intermixing zone and in groundwater springs. The percentage of IoS values increased in the
order glacial effluents > brackish water > stream water > wetland water > lake > groundwater springs
indicating differences in the contribution of recalcitrant DOM.
Assigned compound classes (Fig. 2) give an overview about the composition of DOM. Largest
differences among the sites were found for the aromatic DOM groups. Lake Nam Co had five times



fewer aromatic oxygen (O)-rich molecular classes compared to catchment DOM. For aromatic O-poor
compounds, the lake had almost 20 times fewer compound abundance. For highly unsaturated O-
rich molecular formulae Lake Nam Co showed higher values compared to catchment DOM, while
highly unsaturated O-poor molecular formulae were slightly decreased. Further, the unsaturated O-
poor and unsaturated N-containing DOM classes were higher in lake DOM compared to DOM
sampled in terrestrial systems. Within catchments, highly unsaturated O-rich formulae were more
abundant in glacial influenced DOM of Qugaqie. Significantly more unsaturated O-rich molecular
formulae were found in Niyaqu DOM compared to Qugaqie (p=0.02), while unsaturated O-poor
molecular formulae were significantly less abundant in DOM of Zhagu compared to both Niyaqu and
Qugaqie samples (p=0.007). Relative abundances of saturated molecular formulae were overall low
(Fig. 2).





**Table 1: Overview on mean and standard deviation of indices and elemental composition ratios and mean and standard deviation of molecular formulae counts for sites.**

| Variable | Niyaqu | | Qugaqie | | Zhagu | | Lake Nam Co ‡ |
|---|---|---|---|---|---|---|---|
| | Mean | SD (±) | Mean | SD (±) | Mean | SD (±) | Value |
| $AI_{mod}$ | 0.33 [a] | 0.05 | 0.32 [a] | 0.02 | 0.36 [a] | 0.02 | 0.25 |
| DBE | 9.43 [a] | 1.05 | 9.52 [a] | 0.93 | 10.22 [a] | 0.32 | 8.14 |
| O/C | 0.46 [a] | 0.02 | 0.45 [a] | 0.02 | 0.45 [a] | 0.02 | 0.44 |
| H/C | 1.12 [a] | 0.08 | 1.17 [b] | 0.05 | 1.13 [a,b] | 0.08 | 1.20 |
| CHO | -0.21 [a,b] | 0.11 | -0.26 [b] | 0.08 | -0.17 [a] | 0.04 | -0.29 |
| $I_{Deg}$ | 0.55 [a] | 0.15 | 0.59 [a] | 0.10 | 0.77 [b] | 0.02 | 0.33 |
| $I_{Terr}$ | 0.32 [a] | 0.13 | 0.35 [a] | 0.03 | 0.34 [a] | 0.04 | 0.08 |
| $I_{oS}$ [%] | 14.4 [a] | 1.7 | 14.7 [a] | 1.6 | 18.0 [b] | 2.2 | 17.2 |
| Number of formulae containing N | 1412 [a] (49.2) | 941 | 1928 [a] (54.9) | 1083 | 1933 [a] (56.5) | 951 | 1284 (58.8) |
| Number of formulae containing P | 143 [a] (4.9) | 112 | 129 [a] (3.6) | 96 | 130 [a] (3.8) | 54 | 196 (8.9) |
| Number of formulae containing S | 69 [a] (2.4) | 96 | 146 [b] (4.1) | 101 | 46 [a,b] (1.3) | 43 | 37 (1.6) |
| Total number of molecular formulae | 2867 [a] | 1060 | 3509 [a] | 1340 | 3416 [a] | 848 | 2183 |

*Significant differences are indicated by superscript letters (a,b,c) ‡ indicates sample size n<3, here no standard deviations are given and no statistical tests were performed. For heteroatoms (N, P, S) percentages of the total number of molecular formulae are given in brackets. Boxplots of data are presented in the supplementary materials (Fig. S2).*

**Table 2: Overview on mean and standard deviation of indices and elemental composition ratios and mean and standard deviation of formulae counts for sample categories (including endmembers).**

| Variable | Glacial effluent | | Spring ‡ | Wetland ‡ | Stream water | | Brackish water | | Lake water ‡ |
|---|---|---|---|---|---|---|---|---|---|
| | Mean | SD (±) | Value | Value | Mean | SD (±) | Mean | SD (±) | Value |
| $AI_{mod}$ | 0.31 [a] | 0.04 | 0.35 | 0.33 | 0.33 [a] | 0.05 | 0.35 [a] | 0.02 | 0.25 |
| DBE | 9.15 [a] | 1.06 | 9.76 | 9.70 | 9.51 [a] | 0.97 | 10.39 [a] | 0.73 | 8.14 |
| O/C | 0.45 [a] | 0.03 | 0.44 | 0.46 | 0.46 [a] | 0.02 | 0.46 [a] | 0.01 | 0.44 |
| H/C | 1.17 [a] | 0.08 | 1.15 | 1.09 | 1.13 [a] | 0.07 | 1.13 [a] | 0.07 | 1.20 |
| CHO | -0.28 [a] | 0.14 | -0.18 | -0.19 | -0.22 [a] | 0.09 | -0.18 [a] | 0.07 | -0.29 |
| $I_{Deg}$ | 0.53 [a] | 0.13 | 0.77 | 0.53 | 0.58 [a] | 0.15 | 0.66 [a] | 0.02 | 0.33 |
| $I_{Terr}$ | 0.34 [a] | 0.06 | 0.32 | 0.34 | 0.32 [a] | 0.11 | 0.39 [a] | 0.04 | 0.08 |
| IoS [%] | 13.7 [a] | 2.24 | 17.4 | 16.0 | 15.1 [a] | 1.9 | 14.5 [a] | 1.33 | 17.2 |
| Number of formulae containing N | 1548 [a] (52.2) | 1135 | 1261 (44.7) | 2549 (62.3) | 1511 [a] (44.2) | 922 | 2586 [a] (57.5) | 1231 | 1284 (58.8) |
| Number of formulae containing P | 103 [a] (3.4) | 90 | 117 (4.1) | 291 (7.1) | 130 [a] (3.8) | 91 | 231 [a] (5.1) | 160 | 196 (8.9) |
| Number of formulae containing S | 134 [a] (4.5) | 125 | 9 (0.3) | 68 (1.6) | 77 [a] (2.2) | 90 | 163 [a] (3.6) | 126 | 37 (1.6) |



| Total number of formulae | 2965 [a] | 1132 | 2819 | 4091 | 3416 [a] | 1053 | 4492 [a] | 1639 | 2183 |
|---|---|---|---|---|---|---|---|---|---|

*Significant differences are indicated by superscript letters (a,b,c) ǂ indicates sample size < n=3, here no*
*standard deviations are given and no statistical tests were performed. For heteroatoms (N, P, S)*
*percentages of the total number of molecular formulae is given in brackets. Boxplots of data are*
*presented in the supplementary materials (Fig. S3).*

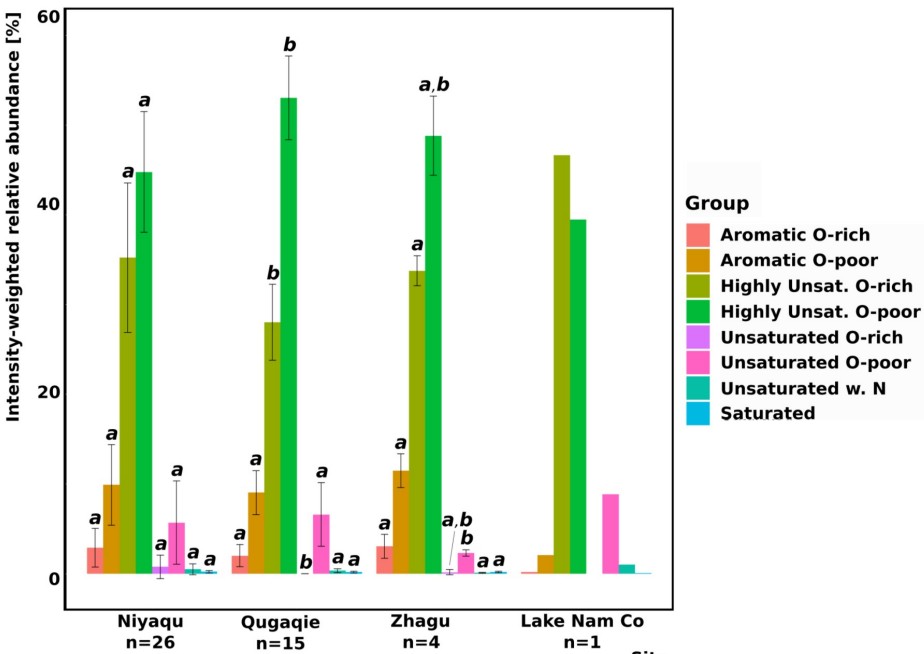


**Figure 2: Mean relative intensity-weighted counts of descriptive compound classes for stream waters of the three catchments and Lake Nam Co. For sample sizes n<3, no standard deviations are given. Error bars indicate standard deviations, significant differences are indicated by superscript letters (a,b,c).**

Lake water samples contained only one fourth of aromatic O-rich compounds compared to stream
water samples. Also, for aromatic O-rich compounds, large differences exist between lake water and
water samples assigned to other sample categories, with samples from Nam Co having smallest
abundances in this class. Brackish waters had significantly more aromatic O-poor compounds compared
to glacial effluents (p=0.05). The relative abundances of highly unsaturated O-poor were highest in
glacial effluents and groundwater spring waters, while wetland waters and samples from Lake Nam Co
had the smallest relative values. Water samples from the Nam Co Lake had the highest relative
abundances of highly unsaturated O-poor DOM compared to all other groups, especially glacial effluents
had 40% less relative abundances in this class.





Unsaturated O-rich compounds had a very low proportion, but were more abundant in stream,
glacial and groundwater compared to wetland, brackish and lake waters, while unsaturated O-poor
compounds were more frequent in lake waters and glacial effluents compared to the other
categories. Spring water exhibited the least relative contribution of unsaturated O-poor compounds,
being almost 20 times lower compared to water from Lake Nam Co. Lake and glacial DOM were rich
in these molecular formulae, followed by stream, brackish and wetland waters which showed 50%
less abundance in unsaturated N-containing formulae compared to lake DOM. Spring water had
fewest of this compound class, being 75% lower compared to Lake Nam Co (Figure 3). Saturated
compounds were overall low, slightly higher values were only encountered in stream, spring, and
brackish waters.

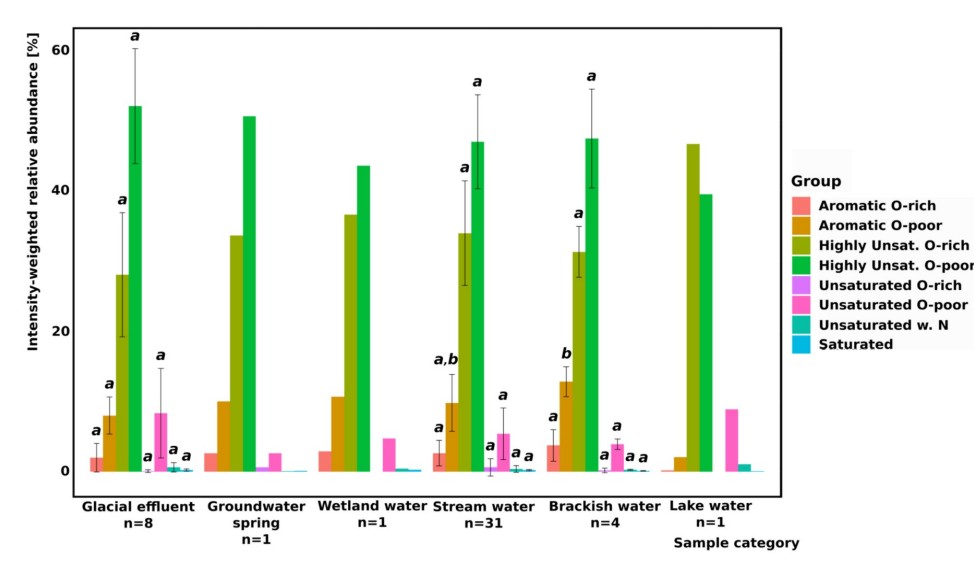

**Figure 3: Mean relative intensity-weighted counts of descriptive compound classes sorted for sample categories. Error bars indicate standard deviations. For sample sizes n<3, no standard deviations are given. Error bars indicate standard deviations, significant differences are indicated by superscript letters (a,b,c).**

3.3 NMDS ordination
NMDS ordination was conducted with an overlay for both independent factors. NMDS stress value of
0.14 is in line with accepted stress measures (< 0.2; Dexter et al., 2018). The NMDS showed an
expansion of molecular formulae in the ordination plane in a funnel-shaped circumference above the
coordinate origin. The clustering of samples was within the scatter of formulae separated in two
smaller groups above the origin of coordination and more closely to the origin (Fig. 4a). $AI_{mod}$ values
increased towards the negative direction of the first NMDS dimension and decreased towards the



positive direction, respectively (Fig. 4a). In the positive direction of dimension 1 of the NMDS biplot
(Fig. 4b), unsaturated and saturated compound classes are resolved together with sulphur for the
internal variables. DIC and EC explained this direction as external variables (black vectors in Fig. 4b).
In the negative direction of dimension 1 aromatic compound classes, the $AI_{mod}$, DBE, $I_{Deg}$ and $I_{Terr}$
indices were loading. For external variables: plant cover, DOC and FDOM and partly $\delta^{13}C$ of DOC were
loading in this direction. The positive direction of dimension 2, distinguished samples with high
abundances in highly unsaturated and saturated compounds, together with the heteroatoms N, P
and S. The external variable of $NO_3$ concentration also loaded in this direction. The negative direction
of dimension 2 clustered samples close to the origin of the ordination which were related to IoS
values. Water samples of the Zhagu catchment were located around the centre of the ordination
plane in figure 4b, while DOM of Qugaqie was placed below and above the centre of the dimension
plane. Water samples from the Niyaqu catchment were more uniformly positioned below the centre.
The sampling categories showed that stream water samples were scattered over a wide plane of the
ordination, while brackish DOM samples were placed in the upper right corner of the plane, except
for one sample with less negative $\delta^{13}C$ of DOC. Glacial DOM clustered around the top-right to
bottom-right half of the ordination, indicating glacial DOM to be associated with heteroatoms,
saturated and unsaturated molecular compounds but as well with the external variables DIC
concentration and less depleted $\delta^{13}C$ of DOC. Glacial DOM signatures from Niyaqu and Qugaqie were
distinct in the ordination.
**4. Discussion**
4. 1 Catchment properties shape DOM composition in the Lake Nam Co watershed
The three investigated catchments of the Nam Co watershed differed significantly in their molecular
DOM composition. The Qugaqie catchment showed the largest number of chemical formulae
identified, together with a higher abundance of formula containing S heteroatoms. The large α-
diversity of Qugaqie DOM can be influenced by the productivity of the glacial ecosystem hotspot
(Hodson et al., 2008) adding to the alpine pasture signature. The strong influence of S heteroatoms is
likely related to the local weathering processes by sulphate oxidation in the Zhadang glacier (Yu et
al., 2021). Anaerobic metabolism of sulphate reducing bacteria is known to take place in glacial
sediments and ice (Wu et al., 2012). After biotic reduction, sulphate is abiotically incorporated into
DOM, forming dissolved organic sulphur compounds (Pohlabeln et al., 2017). Unsaturated O-rich and
highly unsaturated O-rich molecular formulae were relatively depleted compared to the other
catchments, while highly unsaturated O-poor and unsaturated O-poor formulae were increased.
Correspondingly, the CHO index indicated a comparably low degree of C oxidation. This shows that
higher complex compounds originating from plants and soils dominated the DOM of Qugaqie.



Correspondingly, the degree of microbial breakdown was lower compared to the other two
catchments, explaining the larger abundance of O-poor molecular formulae as also found in
incubation studies (D'Andrilli et al., 2019).

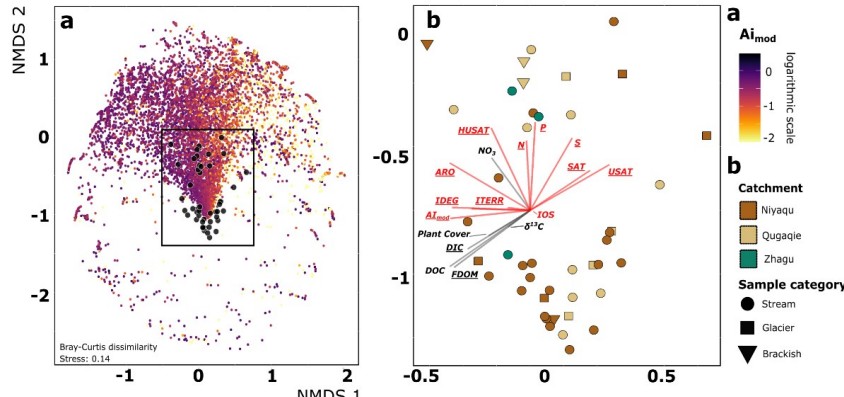


**Figure 4: Non-metric multidimensional (NMDS) scaling for a) samples and elemental composition**
**of compounds (elements C, H, O, N, P and S) plotted with the colour-coded logarithmic scale of the**
**aromaticity index AI$_{mod}$ (Koch and Dittmar 2006; 2016). And b) for samples and environmental**
**variables: internal explanatory variables (red) and external explanatory variables (black).**
**Underscores indicate significant (α = 0.05) relation of environmental parameters with the NMDS**
**dimensions. Stress = 0.14, k = 3, dissimilarity index = Bray-Curtis. ARO= aromatic O-rich and**
**aromatic O-poor, HUSAT =highly unsaturated O-rich and highly unsaturated O-poor, USAT =**
**unsaturated O-rich, unsaturated O-poor and unsaturated with N, SAT = saturated. Sites are**
**represented by the colour of sample points, while sample categories are represented by point**
**shapes.**

The Qugaqie catchment is mostly defined by its steep altitudinal gradient (>2000 m) and its glacial
influence, with glacial meltwater as the dominating water source (Bolch et al., 2010; Gao et al.,
2015), relatively low water temperatures of the meltwater hamper the microbial oxidation of DOM,
while water turbidity and turbulence shields against photooxidation. The higher relative abundance
of unsaturated low-molecular compounds makes microbial DOM derived from the glacial ecosystem
probable (Anesio et al., 2009; Hood et al., 2009; Spencer et al., 2014) and further suggests its
preservation through low water temperatures (Adams et al., 2010).
The contribution of glacial meltwater was smaller in the Niyaqu catchment and absent in Zhagu.
Here, the streams were less turbid and flowing slower (Maurischat et al., 2022). The main water
sources are precipitation and groundwater (Adnan et al., 2019a; Anslan et al., 2020; Tran et al.,
2021). In the arheic Zhagu catchment, a stronger indication of aromaticity with higher AI$_{mod}$ and
higher relative abundance of aromatic compounds was found, alongside with a relatively low
contribution of lower-molecular mass unsaturated compounds. Furthermore, the higher oxidation



state indicated by CHO index (Fig. 5b, Table 2) and higher degradation index (Table 2) suggest an
intense microbial processing of biopolymers. This was also corroborated by the higher share of the
IoS, underlining that relatively more complex highly unsaturated O-rich molecular formulae are left
behind and by this enriched as recalcitrant DOM, as was shown in incubation studies (Mann et al.,
2015). The Niyaqu catchment in comparison showed a high contribution of aromatic compounds
together with a high contribution of unsaturated compounds and low H/C ratios. This catchment is
largely influenced by DOM originating from plants such as lignin and its degradation products
(Roebuck et al., 2018; Seifert et al., 2016), underlining the existence of a terrestrial – fluvial pathway.
In comparison, the high-altitude Qugaqie catchment also comprises of a glacial-borne microbial DOM
source highlighting compositional changes in the glacial – fluvial pathway. All catchments,
irrespective of the composition of the water source show influences of terrestrial allochthonous
organic material in DOM. This is visible by equally high aromatic indices ($AI_{mod}$, DBE, $I_{Terr}$) and similar
contributions of aromatic compound classes. This serves as a strong indication of a land use control
on DOM signatures, namely by the human-induced *K.pygmaea* plagioclimax. There appears to be a
steady input of terrestrial DOM along the streampath from pastoral land as also demonstrated for
other agricultural systems (Roebuck et al., 2018). Notably, this influence became smaller, when
glacial, microbial DOM sources were more dominant on catchment level (Fig. 3b).
**4.2 Abiotic and biotic controls on DOM composition are superimposed by pasture degradation**
Our data show, that molecular diversity and DOM characteristics can be pinpointed to landscape
units and endmembers (summarized in the conceptual model in Fig. 7). NMDS analysis (Fig. 4)
suggested that glacial effluents were diverse in chemical composition, but differed between glacial
ecosystems, as was also shown by Spencer et al. (2014). But generally, glacial DOM from Qugaqie and
Niyaqu contained two different signatures from different sources. These were, 1) high abundances in
unsaturated compounds with and without nitrogen as well as a low oxidation state of carbon (Fig.
5a), high ratios of H/C and lowest percentage of recalcitrant DOM as demonstrated by comparatively
low IoS abundances. These parameters indicated a relatively fresh, reduced (oxygen-poor) DOM of
low-molecular mass derived from autotrophic microbial activity at the partly anoxic ice shield and are
in-line with findings from other glacial environments worldwide (Hood et al., 2009; Telling et al.,
2011; Anesio et al., 2009). 2) aromatic and highly unsaturated compound classes and aromatic and
terrestrial indices ($I_{Terr}$, $AI_{mod}$, DBE) were in-line with other terrestrial DOM sources in our dataset,
despite the absence of plant cover in the glacial zones (Maurischat et al., 2022). Glaciers are
understood to receive compounds with higher molecular mass from aeolian deposition, either
condensed (poly)aromatics, e.g. from the burning of fossil fuels (Takeuchi, 2002) or compounds
uncondensed but rich in phenols, usually associated with vascular plants and soils (Singer et al.,
2012). The investigated coexistence of autochthonous and allochthonous DOM sources in glacial



DOM make them diverse and complex and render the understanding of glacial DOM reactions on climatic changes and its downstream lability difficult (Singer et al., 2012).

Groundwater spring DOM from the upland of the Zhagu catchment mainly comprised aromatic and highly unsaturated DOM compounds. Molecular diversity and the number of N, P and S heteroatoms was low. Concurrently, spring DOM suggested a strong degradation by high $I_{Deg}$ and CHO indices (Fig. 5b) and a large contribution of recalcitrant organic matter as indicated by IoS values. Spring water is generally expected to inherit aquifer and catchment characteristics in its DOM signature, also preserving terrigenous DOM (Osterholz et al., 2022; Yoo et al., 2020). The shallow groundwater table as found in Zhagu (Tran et al., 2021), was shown to be in contact with soil organic matter and yak faeces (Maurischat et al., 2022), both of which can leach soil-borne OM to the groundwater table (Connolly et al., 2020) and re-emerge at groundwater springs. The low molecular diversity of groundwater spring DOM responds to an enrichment in aromatic and highly unsaturated formulae, given the high degradation and recalcitrant nature of DOM, these compounds likely originated from the degraded pedosphere and have been transported with the groundwater. The connection of over-used, degraded pastures of Zhagu (Fig. 1 & 4b) with groundwater spring DOM indicated that a highly modified DOM signature is emitted from springs and retrieved in streams. While investigations of the effects of pasture degradation on Tibetan soil OM stocks are receiving much attention (Liu et al., 2017), degradation induced changes of DOM composition have fallen short. DOM as a marker for *K.pygmaea* degradation should generally receive more attention when studying degradation effects on landscape-scale.

DOM from an extensive alpine wetland of the Niyaqu catchment shows a high α-diversity (>4000 assigned molecular formulae), was rich in N and P heteroatoms and in terrigenous highly unsaturated O-rich and unsaturated O-poor compounds. In a study from the same site, wetland DOM was enriched in mineralized nitrogen and DOC compared to the surrounding streams (Maurischat et al., 2022). It was reported that alpine wetlands are highly productive and contain large amounts of nutrients in the organic biomass and organic matter. Accordingly, the degradation of wetlands can pose a nutrient threat to downstream ecosystems (Bai et al., 2010; Zhang et al., 2020).

Stream samples cluster relative widely around the NMDS ordination space but are concentrated in the lower centre (Fig. 4b). Most samples had a predominance of aromatic compounds either associated with highly unsaturated O-poor or highly unsaturated O-rich formulae, suggesting an input of terrestrial compounds, such as lignin and tannin and their degradation products (Mann et al., 2015), corroborated by depleted $\delta^{13}C$ DOC signatures (Maurischat et al., 2022). The *K. pygmaea* biome spreads as an azonal pasture along the streams (Fig. 1). Roebuck et al. (2020) pointed out that agricultural sites provide terrestrial inputs to surface waters. Correspondingly, Lu et al. (2015) found



predominance of aromatic and highly unsaturated compounds in watersheds dominated by pastoral
activity. The *K. pygmaea* biome is a large alpine yak pasture with potential influence of faeces, known
for their high biolability (Du et al., 2021), suggesting low molecular mass and negative CHO. The
scattering of stream samples can therefore also be explained by changing inputs from the pasture
biome. Faeces input and products of their microbial utilisation, will be associated with increases of N-
containing unsaturated formulae and saturated formulae (Vega et al., 2020). Still, the input from soil
and terrestrial plant material dominated stream samples.

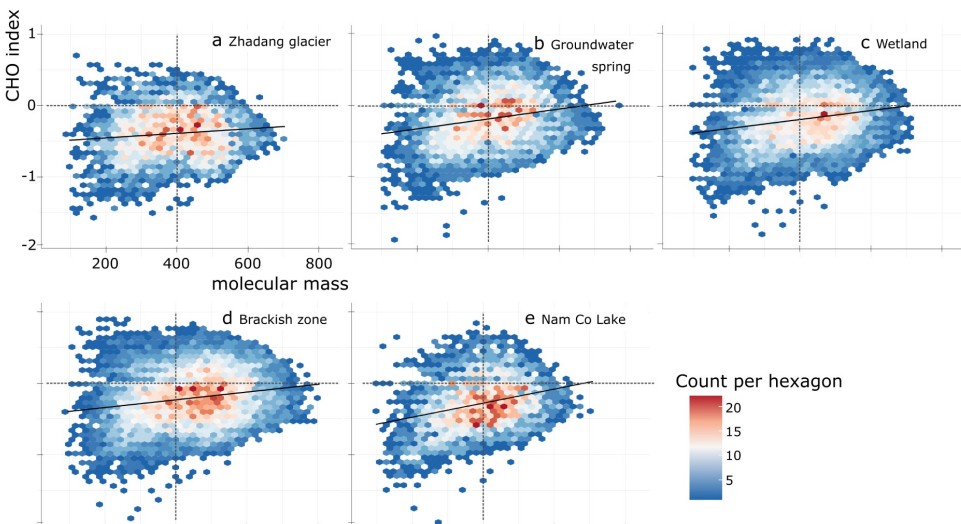


**Figure 5: Hexagon scatter plots show the count of chemical formula in the space of CHO index (Mann et al., 2015) and molecular mass. Black lines represent the linear model of CHO index and molecular mass of the respective sample, grey dashed lines split the plot in quadrants for orientation. Regression and orientation lines are printed to guide the eyes only. a) Ice from a glacier in Qugaqie, b) groundwater spring from the upland of Zhagu (3:10), c) water from the standing water column of a Niyaqu wetland (1:26), d) water from the brackish zone of Lake Nam Co and the Qugaqie stream (2:15) and e) water sample from 30 m depth of the Lake Nam Co. Axis scales are fixed.**

4.3 The Lake reactor: photooxidation changes DOM molecular composition
Brackish water samples showed the overall highest molecular diversity, highest number of N and S
heteroatoms and as well highest number of aromatic O-poor compounds together with high $AI_{mod}$,
$I_{Terr}$ and DBE indices (Tab. 2). These samples were mostly dominated by aromatic and highly
unsaturated O-poor formulae and by this retained and accumulated the terrestrial signal of streams.
Van Dongen et al. (2008) described brackish waters as a zone of gradual mixing. Several studies
found a connection of terrestrial-derived DOM signatures from streams to be exported to marine
systems and showed gradual mixing in the brackish zone (Benner et al., 2004; Dittmar and Kattner,
2003; Pettersson et al., 1997; Ruediger, 2003). More so, the relative increase of aromatic compounds



suggests to some extent a selective degradation and oxidation of lower-molecular mass compounds
in the wash of the waves as suggested by the CHO index (Fig. 5d) and $I_{Deg}$, as shown for estuaries
(Asmala et al., 2014). We suggest that the high molecular diversity in brackish samples represents
both, the terrestrial input from catchment streams mixing with the DOM signature of lake water and
selective degradation of DOM, which is visible for the exemplary mass-to-charge (*m/z*) region
between 565.5 and 567.5 Da when comparing the evolution of molecular compounds from Zhadang
and Qugaqie mid-stream towards its brackish zone (Fig. 6a-c).
Lake water differed in its DOM composition compared to all other samples (summarized in the
conceptual model in Fig 7). Lake DOM was relatively enriched in unsaturated and saturated
compounds, which can include lipids and carbohydrates, but depleted in aromatic and highly
unsaturated O-rich formulae. Correspondingly, $AI_{mod}$, DBE and $I_{Terr}$ decreased, likely as a result of
photooxidation (Helms et al., 2014) given the clear water column and high irradiation (Wang et al.,
2020). Spencer et al. (2009) reported photooxidation to remove aromatic DOM such as lignin phenols
from a large river system and Helms et al. (2014) investigated the loss of DOM optical properties
after light exposition. This process can hence explain the depletion of phenolic constituents
corresponding to decreased $AI_{mod}$, $I_{Terr}$ and DBE indices and becomes visible in the excerpt of mass
spectra between 565.5 to 567.5 Da when comparing brackish and lake DOM (Fig. 6c-6d). In a stream-
estuary-lake gradient the terrigenous brackish DOM likely underwent a transformation when
entering deeper in the lake, leaving more recalcitrant DOM behind (Goldberg et al., 2015) as
corroborated by the higher percentage of IoS compared to stream and brackish DOM (Table 2).
Miranda et al. (2020) found that highly unsaturated and aromatic compounds are not only degraded,
but partly transformed to unsaturated (N-containing) compounds by UV radiation. This can further
explain the increase in unsaturated and unsaturated N-containing compound classes and the more
negative CHO with lower molecular mass (Fig. 5e) in lake DOM (Figure 3). Along with degradation and
transformation of allochthonous DOM, autochthonous DOM production plays a large role for the
natural OM characteristics of Lake Nam Co. Microbial autochthonous DOM sources have been
suggested for Lake Nam Co (Spencer et al., 2014; Maurischat et al., 2022) and other large lakes of the
Himalayas (Liu et al., 2020). Hu et al. (2016) report from a Nam Co food web study that mainly
autochthonous organic carbon sources are utilised by zooplankton, further corroborating the
importance of an autochthonous DOM source. The comparably low CHO index (Fig. 5e) underlines
the existence of autochthonous low molecular mass, reduced carbon species, while higher molecular
masses of allochthonous origin are more oxidised in the lake environment compared to other
systems (Fig. 5), indicative of strong processing. The DOM signature of water samples of Lake Nam Co
from 30 m depth were drawn below the dimictic thermocline (Wang et al., 2009). This DOM
resembled characteristics of the open ocean, being low in aromatic compounds and terrigenous




indices (Seidel et al., 2017) and had a larger percentage of IoS, suggesting higher recalcitrance and
millennial scale stability. Comparably DOM recalcitrance of Lake Nam Co was not on the level of large
arctic rivers (Behnke et al., 2021) or the deep ocean (Lechtenfeld et al., 2014), but clearly increased
compared to the watershed stream network. Lake DOM further exhibited autochthonous DOM
sources derived from algal and microbial production (Zark and Dittmar, 2018; Seidel et al., 2015).
DOM of this large endorheic Tibetan lake is evidently not influenced by DOM of inflowing streams,
since the lake is a functional reactor in processing terrigenous aromatic DOM.

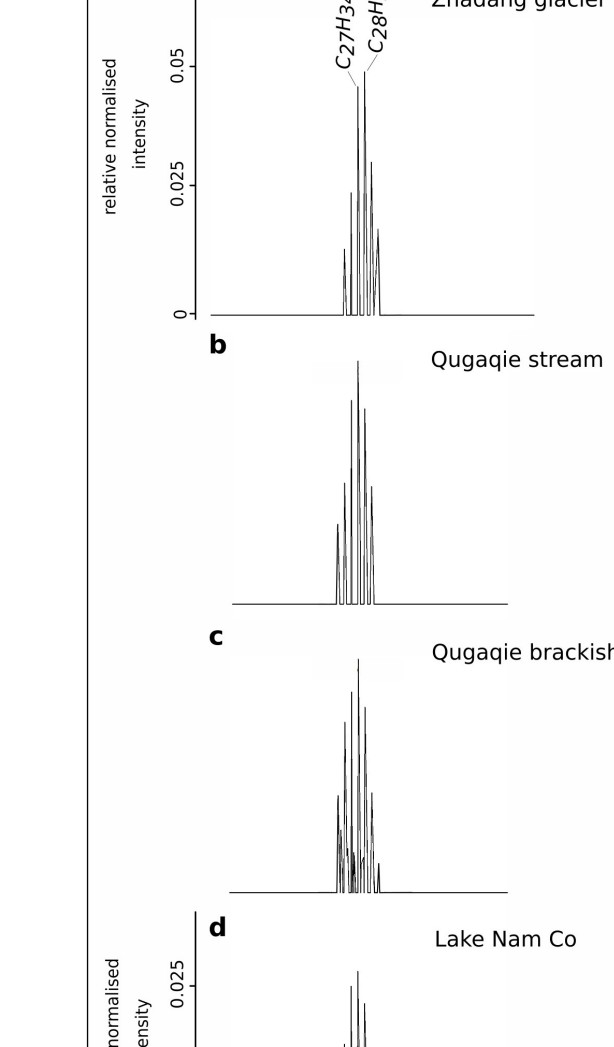

**Figure 6: Molecular cross-section through the Qugaqie catchment with the termination in Lake Nam Co. Presented mass spectra between 565.5 and 567.5 *m/z* are noise reduced and normalised for intensity. A) Ice from a glacier in Qugaqie, b) mid-stream sample from Qugaqie (2:12), c) water from the brackish zone of Lake Nam Co (2:15) and the Qugaqie stream and e) water sample from 30 m depth of the Lake Nam Co. Axis scales are fixed.**



### 5. Conclusions

We elucidated the composition and the processing of DOM along the flow path in the High Asian endorheic Nam Co watershed. We investigated different endmembers of three catchments, including DOM samples of glaciers, groundwater springs, alpine wetlands, streams, the brackish mixing zone and the saline lake.

Catchments of the Nam Co watershed differed in their chemical composition of DOM irrespective of the spatial distances. Conversely, endmembers, and degradation of alpine pastures, were found to influence these differences by releasing degraded terrestrial DOM signatures. Water sources influence DOM signatures in streams of the Lake Nam Co watershed. Molecular diversity was large in glacial influenced streams, which were also characterized by the largest proportion of low molecular mass compounds. In the glaciated Qugaqie catchment, we identified a unique dual source, on the one hand, a microbial, low-molecular mass DOM fraction relatively rich in S heteroatoms and unsaturated compounds with and without nitrogen, possibly including degradation products of peptides and amino sugars, suggesting high biolability and autotrophic DOM production in the glacial ice shield. On the other hand, DOM with high aromaticity and high abundances of highly unsaturated compounds, such as plant-derived lignin, hinting at a depositional source of aeolian transported local dust, derived from soil. Polycondensed aromatics probably including black carbon with high an $AI_{mod}$ and large number of C-atoms derived from the combustion of fossil fuels or household burning of yak dung may also enter the ice shield via the atmospheric pathway. The large influence of glacial meltwater in the Qugaqie catchment greatly modifies DOM signatures along the whole stream of the catchment and probably delivers bio-available compounds to the southern lake shoreline, underlining the existence of a glacial – lacustrine pathway. The Niyaqu and Zhagu catchment comprised a lower molecular diversity, and had a mainly allochthonous DOM source of highly unsaturated and aromatic compounds, attributed to the input of surrounding plants and soils of the pastoral *K.pygmaea* biome to the streams.

Groundwater spring DOM had a low molecular diversity and was enriched in plant and soil-derived aromatic and highly unsaturated compounds alongside with an increase in P heteroatoms and saturated formulae. This suggests that spring DOM of the Zhagu upland constitutes background DOM signatures but also inherits highly degraded, oxidized, and probably recalcitrant DOM from the degrading pastures and potentially yak faeces. Wetland DOM exhibited a high molecular diversity and was especially rich in N-heteroatom compounds, alongside with aromatic and highly unsaturated formulae. Wetland DOM likely represents a broad range of terrestrial DOM signatures from the catchment, driven by the high productivity, water-logging and the basin topography of the wetlands. The latter lead to a steady inflow of water and OM by lateral movement. If the reported degradation



of alpine wetlands in High Asia drives a larger release of DOM to the streams, this can be considered
a threat for oligotrophic terminal aquatic systems, such as Lake Nam Co, but especially for smaller
lakes.
Stream samples were mostly associated with the input of allochthonous materials, originating from
vascular plants and soils. These are attributed to the predominant *K. pygmaea* biome and pastoral
practise stretching along the path of streams. From these zones terrestrial-borne DOM is constantly
refuelled into the streams.
Brackish samples represented the mixing zone of stream and lake water, showing that the terrestrial
DOM signal is being transported with stream waters into the lake. In this cold, fast flowing streams
with short water residence times and high turbidity transformation along the pathway is arguably
smaller compared to other ecosystems. Lake DOM, however, was chemically diverse compared to all
other sources, its molecular composition suggests intense photooxidation and transformation of
imported allochthonous stream DOM, alongside with an autochthonous DOM source from microbial
and algal in-situ production in the large oligotrophic lake. These DOM signatures can be compared to
terminal signatures encountered in the open ocean. Our study shows that DOM cycling in the Nam
Co catchment needs a thorough assessment, since it can be diverse between catchments and
landscape units. In order to safeguard water resources and related ecosystem services, knowledge
about the different sources and their later processing is indispensable. DOM properties have proven
here as a selective proxy, suitable to be implemented as a monitoring agent in the Nam Co
watershed, representative for processes on the larger southern TP.



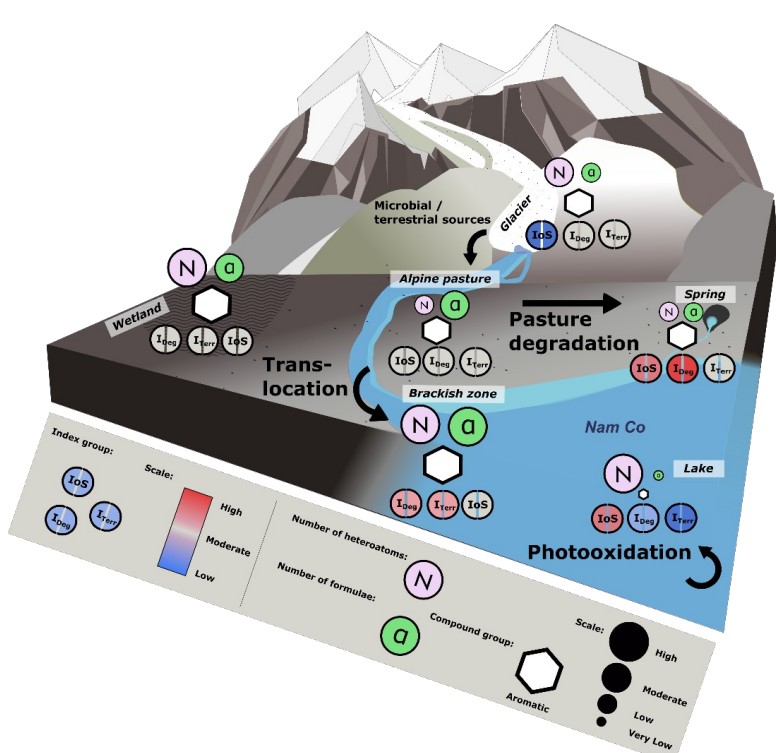


**Figure 7: Overview of the molecular DOM information along the water network continuum of the TP. The number of N-heteroatoms are indicated relative to the total number of molecular formulae. Island of stability (IoS), degradation index ($I_{Deg}$) and terrestrial index ($I_{Terr}$) are ranging in a red colour code for high blue in the opposite case. Alpine steppe and groundwater spring samples are spatially correlated, and thus subject of a common evaluation. Main processes and sources are added and indicated by black arrows.**

## Acknowledgements

The authors thank the staff of ITP-CAS and the NAMORS research station for their hospitality and assistance during the sampling campaign. We further thank the associated members of the TransTiP team for organizing the field work and helping hands during sampling.

This research is a contribution to the International Research Training Group "Geo-ecosystems in transition on the Tibetan Plateau (TransTiP)", funded by Deutsche Forschungsgemeinschaft (DFG grant 317513741 / GRK 2309).

M. Seidel is grateful for DFG funding within the Cluster of Excellence EXC 2077 "The Ocean Floor – Earth's Uncharted Interface" (Project number 390741603).




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
