# Peer review of "Complex dissolved organic matter on the roof of the world Tibetan DOM"

_EGUsphere, 2022_

## Author Response (AR1)

Public justification (visible to the public if the article is accepted and published):

Dear Authors:

Thank you for your detailed responses to the constructive comments and suggestions offered by the5 two referees.

While both reviewers recognized the novelty of your findings, they also raised serious concerns about your data presentations and interpretations. Given the reviewers' overall positive evaluation overshadowed by many critical issues that would require considerable efforts to address, I thought that major revisions would be required before considering the revised manuscript for publication in

Biogoesciences. For your revision, please also take into consideration the following suggestions:

Reply: Thank you for giving us the chance to modify the manuscript and for pointing out these flaws.

I allowed myself to answer your addressed points directly in your reply and marked this here by blue text.

- As both reviewers pointed out, many parts of the manuscript, the abstract and conclusions sections in particular, are (often unnecessarily) too long.

In the case of abstract, the first introductory sentences (L 25-34) can be shortened to leave only a few sentences introducing background information.

Conclusions can also be shortened considerably by focusing on key messages you want to highlight, not repeating key findings.

- As the first reviewer suggested, you need to provide definitions of terms that are not widely used.

For instance, I wondered if 'DOM continuum' in the title actually relates to what you talk about some

'discontinuities' in DOM composition along the hydrologic continuum.

Reply: We agree. We changed the title accordingly to point out complex DOM interaction also with discontinuities along the aquatic continuum. The new title reads: "Complex dissolved organic matter on the roof of the world – Tibetan DOM molecular characteristics track sources, land use effects, and processing along the fluvial-limnic continuum"

In addition, I would suggest that you double check the correct or consistent use of many technical terms throughout the manuscript. In L36, for example, I would opt for "wetland ecosystems", not

"wetland biomes". As you know, biomes usually refer to a regional- to continental-scale ecological32 unit.

Reply: "Biome" was changed to "ecosystem" throughout the MS. 8 replacements were made throughout. Further technical terms were screened for accuracy. Replacements are marked in the text with tracked changes.

You use watersheds and catchments interchangeably, but a consistent use of one term would be37 more "reader-friendly".

Reply: Our intention was to use "catchment" when we refer to the whole Nam Co lake catchment and "watershed" when we refer to the individual subcatchments. We replaced "watershed" by

"subcatchment" throughout the text and now refer to the Nam Co catchment when addressing the whole system.

- Given the significance of DOM quantification and characterization in your study, more quantitative

QC information would help readers assess the overall analytical performance. For instance, how accurate the analysis of reference materials (also used for DOC?) was; how precise the analysis of replicate DOC analysis was,,,

Reply: We addressed the issue of limited QC data in our MS. Adjustments were made in sections 2.2

(blank and reference material standards) and 2.3 (FT analysis). In the *Results* section we expand on the issue under point 3.1 starting from line 254 f. An additional figure was added to the supplementary material (Figure S1) showing the consistency of in-line standards. We feel this is sufficient evidence for the soundness of our technical application.

- Regarding you emphasis on photooxidation, I also think that your descriptions should be more 51 balanced, based on your own findings and comparisons with other studies conducted in similar 52 53 alpine or high-latitude systems. You suggested photooxidation as the most probable mechanism for less aromatic DOM signatures you found in Nam Co. To be able to single out photooxidation from 54 competing processes including biodegradation, I would suggest a more comparative approach that 55 56 could tell readers, for example, why photooxidation overrides potential microbial activities in the specific case of Nam Con. I would appreciate more in-depth discussion of the muti-faceted topic 57 58 (particularly in the section L 537-571). This discussion can correct somewhat misleading statement like "DOM of this large endorheic Tibetan lake is evidently not influenced by DOM of inflowing 59 streams, since the lake is a functional reactor in processing terrigenous aromatic DOM.", I hope. 60

Reply: We tackled this issue from two sites. First, we softened our language and rephrased all sections that were marked as incomprehensible or far-fetched by the reviewers and by you. Second and more profound: We revised the full paper and rewrote wide parts of it. For the specific example

- 64 of photooxidation > microbial utilisation, we revised Figure 6: now it becomes indicative, that
- 65 specifically aromatic, higher molecular weight compounds were removed from the lake water. In the
- 66 discussion of these results we make the mechanistic understanding of this selective removal clearer:
- 67 From the current knowledge of OM transformation it seems unlikely that this removal was caused by
- 68 microbial utilisation of inflowing terrestrial DOM alone. This also fits to the clear lake water and the
- 69 high solar irradiation of the study site. However, we do acknowledge, that indications in DOM of
- 70 microbial activities exist in lake DOM and this was taken care of in the discussion from line 487 f...
- 71 I would like to ask you to make all the changes easily identifiable in a marked-up manuscript based
- 72 on your point-by-point responses to the reviewer comments. If possible, please add up your
- 73 responses to the original reviewer comments (your first responses are very difficult to grasp the
- 74 points quickly) and specify the line numbers of the revised parts in your final responses
- 75 accompanying the revised manuscript.

**Preface**

This is an updated version of our response ex post to the editor's decision. Lines and changes that were requested are indicated here when applicable.

As authors of the manuscript egusphere-2022-1375, titled: "A DOM continuum from the roof of the world - Tibetan molecular dissolved organic matter characteristics track sources, land use effects, and processing along the fluvial-limnic pathway" NOW: "Complex dissolved organic matter on the roof of the world - Tibetan DOM molecular characteristics track sources, land use effects, and processing along the fluvial-limnic continuum", we wish to express our gratitude to the anonymous reviewer No.I for thorough check and productive comments.

**General Statement:**

We will revise the manuscript to be more specific about the "central theme" missed. Indeed, we will highlight the differences along the fluvial-limnic continuum with more respect to endmembers as one part of the storyline (I) and transformation as the other part of the storyline (II).

As of Figure 6; from line 521 to 572 we dedicate a longer paragraph about the lake reactor and its transformations. Table 2 and Figures 3, 5 and 6 document parts of the transformation effects. However, we do see that the information should be better managed in the respective paragraph (i.e. 4.3) and added additional visualization as an "van Krevelen loss plots". Figure 6 shall be understood as an exemplary spotlight on one mass as a *pars pro toto* visualisation.

Detailed comments:

Abstract general comments: we will condense the abstract 26: We prefer to keep this statement because it is directly linked to the motivation and the research question. Pasture degradation is in part a climate change manifestation.

27: You are right, the comparison between glaciated and non-glaciated watersheds did highlight the specificity of glaciation as a determining factor. We will rephrase here. *Line:27; Change: "Carbon cycling on the TP is influenced by glaciation and degradation of the pasture ecosystem."*

30: Will be removed and rephrased

31: We will condense here

33: We will rephrase this and argue with the fluvial-limnic pathway that is already used in the title of the MS Change: pathway was removed we now use "aquatic continuum" throughout when referring to the whole system of glacier-stream-lake

33: Since degradation drivers are mixed and still under debate we will change this to more defined vocabulary *Change:" land-use and climate change"*

38: We will condense the introductory part of the Abstract, however not all readership can be considered to be well acquainted with the natural settings and challenges of High Asia, so we argue that some introduction is necessary. There is debate if "endmember" is a standing term in hydrology, we will nevertheless explain it. The outcome that DOM differs is maybe expected, but we refrain from calling that weak. There was not much research done on High Asian DOM; and DOC loads are low, we think it is a strong finding that so much diversity is carried in DOM.

39-49: Will be rewritten Change: we rephrased in accordance with the new findings for glacial systems, wetlands, the influence of degradation and the lake reactor

50: Thanks, we will put this in the forefront of the abstract Line:35f; Change: We moved this part up and removed duplicate statements

54-56: We will rephrase and keep this as a final statement *Change: we eased the language and made our points more clear: a) diverse DOM in high altitudes b) can* be pinpointed to landscape processes, land-use change, and ecosystems c) can be changed by influences on these landscape processes, land-use change, and ecosystems

57: The definition of how water quality exactly is measured is of course open. DOM is an important part of the terrestrial-aquatic carbon and nutrient cycle and as this of OM cycling. We think that all changes in the riparian interface will impact on DOM characteristics. We will rephrase the sentence to reduce ambiguity. Change: "The close link of alpine SPE-DOM composition to landscape properties is indicative for a strong susceptibility of DOM characteristics to climatic and land use changes in High Asia." In active voice.

59: We will add e.g. "DOM photooxidation" to state that it is a process and add graphical elements to distinguish from "pasture degradation" Change: Please see graphical abstract

**Introduction**

62-71: We will shorten this paragraph, however we argue that DOM and its characteristics translocated with the freshwater will be affected be pasture degradation (as we have corroborated) and that this in turn can affect freshwater quality.

69: peak-water is a standing term in socio-ecology and deeply related to "fossil" glacial water sources. Will be removed

70: Will be removed

74: Will be condensed to the end of the Introduction section Now at lines: 90 f.

74-76: Will be changed; most likely to indicators/ indices. We will introduce the concept of a molecular fingerprint which is better depicting our experimental setup. The word "marker" will be removed throughout the manuscript No further concept (also including "molecular fingerprints") was introduced. We feel that the introduction of another concept will not benefit the MS because of an overload and the need to further define the concepts boundaries. We removed "markers" and "proxies".

76-79: We will rephrase this accordingly and remove the "marker function"; We argue that the comparison of two glaciated watersheds, and one non-glaciated watershed which is highly degraded, is a functional comparison of anthropogenically altered vs. more natural systems. *Change: the sentence was condensed with other statements in this section and lastly removed due to duplication of statements*.

85-86: Will be deleted.

91-99: We think that a general geographical introduction and an introduction into the study object (DOM) should prequel this section. We will expand on the introduction of the *Kobresia pygmaea* pastures and the Nam Co lake.

102-118: Both questions are indeed large and they represent two major research gaps. From this we deduct four hypotheses. We would like to keep four hypotheses, but we will rephrase the hypothesis since they appear not to be straightforward enough. We will formulate hypothesis that can be answered with "yes" or "no" and this will also imply clear statistical evidence, which we will present. In this observational study we will not be able to test single effects, but we present shortened hypothesis:

Change:

1) Investigated subcatchments of Lake Nam Co differ in their molecular composition of DOM. This hypothesis was removed since it was auxiliary to use different sites/subcatchments.

I) SPE-DOM derived from different ecosystems (glaciers, groundwater springs and wetlands) and streams in degraded land possess unique DOM signatures compared to the integrated DOM of subcatchment streams.

II) The SPE-DOM transformation along the stream path is limited, no major compositional shift is expected in-stream.

III) The SPE-DOM characteristics of lake water are chemically distinct from the terrestrial DOM sources and integrated stream SPE-DOM composition.

111: We perform a carry-over of the respective definition from the literature we rely on here. We will make that clear. We will omit to state some formulae as being intrinsically stable. We will identify that recalcitrant in our study always means empirical stability in oceanic contexts under certain environmental conditions. Also pointing to the fact that a change in condition will lead to different biogeochemical outcomes. Change: At this point of the manuscript "recalcitrant" was deleted in order to meet the request for shortened hypothesis. A definition for recalcitrant DOM (vulgo: refractory DOM) is given now in line 208 f.

117-118: We will broaden the wording of this hypothesis. In general, we expect here an immense shift in DOM characteristics between a large endorheic lake and its terrestrial tributaries. There is indeed +20yrs of evidence that DOM of large lakes can differ from tributary DOM (https://link.springer.com/chapter/10.1007/978-3-662-03736-2\_5), this makes it plausible to come up with this hypothesis ex ante. We will improve the hypothesis according to the comment in line 102-108 (see above; yes vs. no + clear evidence).

121-141: In this study we follow a description-scheme from hilltop to lake, we think it is easier this way, since it follows the hydrological cycle.

Figure 1: Wetland water will be highlighted by a halo. Figure 1 d(c) -> bottom left will be replaced by a more straightforward version. *Change: Line 147 f*

274-277: Thanks 🕲 additional QC on in-line drift and blanks was also added here upon editors request

279-367: We will go through the section and thoroughly revise the content. Section will be shortened.

- Figure 2: We used the Viridis colour palette of R that ensures good readability also for larger amounts of classes; after double check with the colour blind tool provided by EGU we found the combination of colours in connection with the clear orientation of bars to suffice for this issue. We will add descriptive information to the x axis labels and populate this through the text where possible and useful (i.e. once per paragraph) Change: The descriptive labels were also used for Figure 4

367: If the reviewer does not object, we would like to keep the right end label of the X-axis, to populate "sample category" throughout the text.

369-393: In section 2.5 we describe the application of external environmental predictor variables that were obtained from a prior investigation. For the external variables, we will introduce these variables so that the readership is not left empty-handed. The internal predictor variables are extensively described in this text, under 2.4. Change: Section 2.5 was moved to the space of section 2.6 for the reason that external predictor variable should be explained after NMDS.

408: We agree that this section needs clarification, and we will improve it. For clarification: Indeed, the main point is that DOM of Qugaqie (strongly glacialfed system) is influenced by terrestrial DOM to a large extent, visible e.g. by  $I_{Terr}$  values (Table 1). Nevertheless, the negative CHO and the composition-shift towards larger relative contribution of O-poor compounds indicate a lower microbial breakdown of terrestrial-borne compounds, corroborated by D'Andrilli (2019): In their study, mostly O-rich molecular formulae were produced after incubation of DOM substrates, while substrates initially had more O-poor formulae

413-422: Spacing will be corrected.

427: Right, we were missing "mass", sorry

439: We do share your understanding of recalcitrance as a non-functional concept Especially when the impression is evoked that certain components are ultimately stable. We will disclaim the concept of recalcitrance more clearly as indicated in the comments of line 111 see line 208 f. in the MS and our marking at point 111 in this document. The context in which we use refractory is that of Lechtenfeld et al.'s study (2014) in a very narrow scope of aquatic (oceanic) systems.

446-449: We will further elucidate on this and we think that the terminology of land-use control is not as suitable as e.g. "influenced by land use". Further, we will revise the MS to increase connection between results and discussion.

449: We will remove the expression.

483: The indicated in comments 111 and 439 see line 208 f. in the MS and our marking at point 111 in this document

488-490: Under this definition our study setup does not support a marker application indeed, because we cannot control all (or even most) of the external effects. We will refrain from the marker wording and rephrase accordingly.

497: We will further clarify the mechanisms of wetland degradation and how they can become endangering for downstream ecosystems by nutrient subsidies. *Change:* we added two publications that complement the topic of wetland degradation and threats:

- Zhang, Y., Wang, G. & Wang, Y. Changes in alpine wetland ecosystems of the Qinghai-Tibetan plateau from 1967 to 2004. *Environ Monit Assess* 180, 189-199 (2011). https://doi.org/10.1007/s10661-010-1781-0
- Gao, J. (2016). Wetland and Its Degradation in the Yellow River Source Zone. In: Brierley, G., Li, X., Cullum, C., Gao, J. (eds) Landscape and Ecosystem Diversity, Dynamics and Management in the Yellow River Source Zone. Springer Geography. Springer, Cham. https://doi.org/10.1007/978-3-319-30475-5 10

498: We will use "cumulate" or "group together" instead.

512: We will enhance the visibility of ledger and regression lines, but we would like to keep the figure in its current state apart from that. Change: The new ledger and regression lines allow a clearer impression of quadrants and trends and by this show how CHO indices change between sampling sites in respect to formulae m/z.

535: connected to comment of "General statement": This range is a representative selection of a relative high molecular mass area (max. was 2000 Da). This area is representative of the processes that DOM undergoes in the lake reactor especially for aromatic compound groups. We will clarify and elucidate on this and we will prepare a van-Krevelen plot highlighting the diminished and disappeared molecular formulae when comparing lake DOM to stream/brackish DOM. We will further clarify that Figure 6 is a representation of the results already discussed. We will further zoom into a narrower nominal mass areas and clearly depict formulae, so that readability is increased for this figure.

545: Will be clarified in accordance with 535

564: Will be removed

565-566:" millennial-scale" will be removed

566: We will remove the citations and statements concerning arctic rivers and open ocean systems. To our knowledge there is only limited application of FT ICR MS in

High Asia so far, but we will inquire for systems that are more suited to be compared (e.g. Lake Qinghai https://doi.org/10.1021/acs.est.0c01681; or other alpine lakes)

570-571: You are right that with the 30 m sample we see very processed DOM compared to watershed influences. We will rephrase the statement to what it is, i.e. we see altered and differing signatures in lake DOM compared to streams asf., which likely result from processing in the lake.

To state a lake uninfluenced from inflows we would need a time perspective that we currently do not have, so we corroborate your statement.

573-626: We will make the conclusions more concise and synchronize with the revised abstract.

609: You are right, should not be stated here in the conclusion; Will be removed

621: All "open-ocean" comparisons will be removed throughout manuscript

625: We will remove all these statements throughout the manuscript and name this "a high-resolution investigation"

626: In this study we have, in large parts, identified the effects of the local sources on stream and lake DOM characteristics. From this aspect the Nam Co watershed can still stand as a case study representing the general inventory and processes of wider parts of the southern TP which have comparable natural features. We will rephrase this part.

**Preface**

This is an updated version of our response ex post to the editor's decision. Lines and changes that were requested are indicated here when applicable.

As authors of the manuscript egusphere-2022-1375, titled: "A DOM continuum from the roof of the world - Tibetan molecular dissolved organic matter characteristics track sources, land use effects, and processing along the fluvial-limnic pathway", we wish to express our gratitude to the anonymous reviewer No.II for thorough check and productive comments.

**Major point:**

First: Falsifiability of Hypothesis 2: Indeed, it is likely that different systems or endmembers have different compositions, but there is no general reason to believe that they are always so. We are convinced that the hypothesis could be falsified. Since there is no knowledge about the degree of this difference that we elucidate and hypothesize here, this first observational study is justified and for us there was no sound scientific basis to infer more in-depth hypothesis a priori.

To answer your comment: We will revise all hypothesis so that they can be clearly answered with a "yes" or "no". The hypothesis will also imply a clear statistical evidence for the answer, we will clearly present this evidence in the MS.

Second: Single samples (N=1) of course are a problem but are here given by the wish not to conduct pseudo-replication and the fact that the catchments did not allow for a higher sample size. Jurisdictional limitations further prevented us from prolonged travels to other systems. We think that the current data scarcity of the TP is also reflected by this and is an inherent problem to address but also to bear. Here we present a first approach that of course can and should be expanded. We like to draw the attention to some of the view other studies of the ͲP and the sample sizes used therein (e.g.: https://doi.org/10.1016/j.gca.2014.08.006 Ν = 6; https://doi.org/10.1016/j.scitotenv.2021.148376 N = 6)

Statistical exclusion of N=1 groups: We will make sure that these flaws will be removed.

Generally: "Careful reassessment of the claims and/or the storyline" We will review the manuscript, given the limitations that we took care to name we.

Hypothesis 3 and 4: Taking a lake sample unfortunately requires a massive logistical effort. This is why we cannot present more - and also why other haven't done so before. A second lake surface sample didn't meet our strict data quality criteria. The question of misrepresentation is a statistical one that future research will have to address. Hypothesis 3 focuses on DOM transformation in the stream, while H 4 focuses on Lake DOM. We will condense the hypothesis to prevent redundancy.

For in stream transformation we excluded all other samples and just analysed stream samples, so we conducted a within-group comparison also with respect to the distance from the source, but there was no significant effect. Residence times are not well reflected by our study, because we do not have flow metering or gauging data to build on. We will take care to make sure to the readership that we have excluded non-stream samples for the analysis of H3, you are right that this is blurred by the unclear hypothesis formulation and the mix-up with H4.

Selective proxy: We will rephrase the selective proxy formulation. In the DOM samples, we found a strong interconnection of the riparian zone that makes a clear indication that the DOM of the degraded Zhagu system is tremendously differing from the other two systems which have more intact pastures. We found sound ground to believe that these changes are not just by chance but driven by the ecological condition. But we agree that the use of words, like: "proxy" or "marker" is not proper in this case. We will omit using these terms.

L213: We will rephrase it to "local peak intensity sum"

L242: We will remove the percentage expression

L264: we will add the statistical linear limits that we used. Change: Pearson correlation coefficient: |r| > 0.75

L279: We wanted to re-state this expression here, in case a reader will skip the "Materials&Methods section". We will review whether this is necessary and then limit it to the M&Ms

L310: We will take care to harmonize the notation

L321: We will express it in "percent-loss"

Table1: We will add sampling sizes here, but we would like to keep them in the figures also for transparency

Figure5: We will add axis denominations but axis labels would be limited to the top-right element *Change: Figure 5 was updated as proposed*

L534-Figure 6: We propose an additional van-Krevelen loss plot where we mark changes in formulae intensity between the brackish sample and the lake samples in greater detail and resolution. We will further zoom into the nominal mass range more deeply and also show single sum formulae. By this we will show the overall difference between the samples and also an in-depth detail of the same thing. Change: In line 472 selective degradation was removed here. The main aspect here will be on differences between lake and brackish DOM to fortify photo-degradation and bio-degradation processes. A study of selective degradation between stream and brackish DOM should have a larger temporal and spatial resolution.

Figure 7: Pasture degradation will be changed to make clear that this is an environmental process. While the other three arrows indicate DOM processes found in our study. We will revise this.

L616: Correct, unfortunately we could not measure flow rates or gauge streams. Initial data were presented by Keil et al. (10.1016/j.quaint.2009.02.022): indicating large discharge rates in Qugaqie, especially due to glacial melt. We currently investigate CO2 emission from stream water of Qugaqie and found a net uptake of CO2 due to abiotic weathering in the stream water, indicating low microbial in-stream activity (under review https://papers.ssrn.com/sol3/papers.cfm?abstract id=4257918 ). We would hypothesize that sorption and desorption, or sorptive fractionation will be limited due to confined availability of exchange sites and substrates. We found overall low DOC (https://doi.org/10.1016/j.scitotenv.2022.156542), unlike in other more active systems (https://doi.org/10.4319/lo.2001.46.8.1921). However, this topic is open to investigation and interesting to tackle for future research.

L570: We will rephrase this and also restrict this expression with regard to the limited data we operate with.

L605: We found a high number of molecular formulae in wetland DOM with (broad range of terr. DOM), high IoS and high percentage of number of N compounds and P compounds (indicating high-productivity - also verified by Maurischat et al. (2022) (Figure 1) and stated in the text in L 495). Water logging further is a necessity to have a wetland and can be tracked by biota at this site (Maurischat et al. 2022) while the basin topography can be inferred from Figure 1. The claims made in this line are not really revolutionary, we will try to rephrase this sentence to make it more understandable. *Change: Additional to this, the section about wetland DOM characteristics (4.2) was reviewed so that the conclusion is more accessible.*

Manuscript egusphere-2022-1375, titled: "A DOM continuum from the roof of the world – Tibetan molecular dissolved organic matter characteristics track sources, land use effects, and processing along the fluvial-limnic pathway"

| Table 1: Overview over the word count of manuscript sections and percent of words removed after |
|-------------------------------------------------------------------------------------------------|
| major revision (resubmit on 24.05.2023)                                                         |

| Word count of: | Submitted original draft | Re-submit after major | Percent removed |
|----------------|--------------------------|-----------------------|-----------------|
|                |                          | revision              |                 |
| Abstract       | 456                      | 357                   | 22              |
| Introduction   | 804                      | 676                   | 16              |
| Materials and  | 2120                     | 2123                  | 0               |
| methods        |                          |                       | (new QC text    |
|                |                          |                       | included)       |
| Results        | 2010                     | 1775                  | 12              |
| Discussion     | 2479                     | 2218                  | 11              |
| Conclusion     | 711                      | 504                   | 30              |
| Total          | 8580                     | 7653                  | 11              |

---

## Author Response (AR2)

Preface

As authors of the manuscript egusphere-2022-1375, we wish to express our gratitude to the anonymous reviewer No.I and the Editor Ji-Hyung Park for thorough check and productive comments.

Editors marked changes:

*"Thank you for your thorough revision incorporating all the comments and suggestions offered during the first review round. One of the same reviewers evaluated the revised manuscript very positively and I also agree with the reviewer. I am pleased to let you know that your manuscript can be accepted for publication after technical corrections of the following editorial points:*

*- The new title (Complex dissolved organic matter on the roof of the world – Tibetan DOM molecular characteristics track sources, land use effects, and processing along the fluvial-limnic continuum): DOM molecular characteristics can be used to track sources…., but I wondered if they could themselves track sources…. Please clarify this grammatical issue and think about an alternative expression such as "Tibetan DOM molecular characteristics indicating sources…".*
*- Abstract: The SPE-DOM rich in aromatic and highly unsaturated formulae visible in the brackish zone of the lake shore contrasted sharply with "that of" the lake samples.*
*- Introduction: The five paragraphs can be better rearranged to enhance the coherency of the paragraphs. I provide here an example: Paragraph 2 ending "But how the DOM characteristics in alpine aquatic systems are influenced by different ecosystems and how DOM responds to ecosystem degradation is not well understood yet."; followed by Paragraphs 1 on TP and 4 on Nam Co; followed by Paragraphs 3 & 5 as the last paragraph on study objectives."*

Detailed comments:

*1. Title: The Title was corrected to: "Complex dissolved organic matter on the roof of the world – Tibetan DOM molecular characteristics indicate sources, land use effects, and processing along the fluvial-limnic continuum"*

2. Wording in abstract: Change was made as suggested

3. Organization of introduction: Change was made as suggested, the marked sentence was replaced.

4. Few minor points to enhance readability:
   a. In line 123: "better understand" was replaced by "assess"
   b. In line 124: "assess" was replaced by "link"
   c. In line 124: "sensitive" was added to attribute inherent value of high-alpine ecosystems
   d. In line 125: "ongoing" was added prior to anthropogenic changes, underlining the pressing issue of human-made change